# Neonatal Hypoglycemia and Neurodevelopmental Outcomes—An Updated Systematic Review and Meta-Analysis

**DOI:** 10.3390/life14121618

**Published:** 2024-12-06

**Authors:** Shivashankar Diggikar, Paula Trif, Diana Mudura, Arun Prasath, Jan Mazela, Maria Livia Ognean, Boris W. Kramer, Radu Galis

**Affiliations:** 1Department of Pediatrics, Oyster Women and Child Hospital, Bengaluru 560043, India; 2Department of Neonatology, Emergency County Hospital Bihor, 410167 Oradea, Romania; 3Doctoral School of Biomedical Sciences, Oradea University, 410087 Oradea, Romania; 4Department of Neonatology, University of Texas Southwestern, Dallas, TX 75390, USA; 5Department of Neonatology, Poznan University of Medical Sciences, 61-701 Poznan, Poland; 6Faculty of Medicine, Lucian Blaga University, 550024 Sibiu, Romania; 7Department of Neonatology, Clinical County Emergency Hospital, 550245 Sibiu, Romania; 8Department of Medical Sciences, Faculty of Medicine and Pharmacy, Oradea University, 410087 Oradea, Romania

**Keywords:** neonatal hypoglycemia, neurodevelopment, brain damage, outcomes, glucose metabolism

## Abstract

Background and Objective: The effects of neonatal hypoglycemia on the developing brain are well known, resulting in poor neurological outcomes. We aimed to perform an updated meta-analysis on neonatal hypoglycemia, the severity of hypoglycemia, and the associated neurodevelopmental outcomes from infancy to adulthood. Methods: A systematic literature search was conducted from inception until March 2024, using the PubMed, CINAHL, Embase, and the CENTRAL databases. Randomized/quasi-randomized trials and observational studies that evaluated at least one of the pre-specified outcomes were included. A random-effects model meta-analysis was performed to yield the pooled OR and its 95% CI for each outcome due to the expected heterogeneity in the studies. The study findings were reported as per the PRISMA guidelines. Neurodevelopmental impairment (NDI), cognitive impairment, and visual-motor or visual impairment were the primary outcomes. Results: A total of 17 studies (19 publications) were included in the final analysis. NDI, as defined by authors, was significantly higher in early- (OR = 1.16; 95% CI = 1.11–1.43) and mid-childhood (OR = 3.67; 95%CI = 1.07–12.2) in infants with neonatal hypoglycemia. ‘Any cognitive impairment’ was significantly more common in infants with neonatal hypoglycemia (OR = 2.12; 95%CI = 1.79–2.52). Visual-motor impairment (OR = 3.33; 95%CI = 1.14–9.72) and executive dysfunction (OR = 1.99; 95%CI = 1.36–2.91) were also more common in the hypoglycemic group. No difference in the incidence of epilepsy, motor impairment, emotional-behavioral problems, or hearing impairment were noted. Certainty of evidence was adjudged as ‘low’ to ‘very low’ for most outcomes. The severity of hypoglycemia was studied at different intervals, with NDI more common with a blood glucose interval between 20 and 34 mg/dL (1.1–1.9 mmol/L). Conclusions: Low-quality evidence from large observational studies finds a significant association with hypoglycemia in the early neonatal period and long-term neurodevelopmental problems. Additional studies with long enough follow-up are paramount to determine the cut-off concentration and to quantify the impact beyond the infancy period.

## 1. Introduction

Hypoglycemia is the most common metabolic condition, occurring in four in ten infants in the neonatal period [1,2]. It is one of the most easily preventable causes of adverse long-term neurodevelopmental outcomes. An abrupt cut-off from continuous maternal glucose after birth predisposes newborns to transient hypoglycemia as part of their physiological metabolic adaptation during the first 24–48 h of life [3]. Most of the neonates are asymptomatic as the inherent mechanism of counter-regulatory hormones and ketones protect the developing brain from damage [3,4]. In a few of them, including those with risk factors leading to high needs of glucose, insufficient glycogen stores or sickness, hypoglycemia predisposes infants to long-term neurological damage if not detected and treated appropriately [4]. Neonatal hypoglycemic encephalopathy or neonatal hypoglycemic brain injury (NHBI) is used to refer this clinical situation, which is very inhomogeneous [5]. The long-term consequences of neonatal hypoglycemia have been a subject of increasing interest in research. Severe and persistent hypoglycemia has been linked to neuronal death in specific brain regions, including the cerebral cortex, basal ganglia, and the parieto-occipital region [6,7]. The pathogenesis involves mechanisms such as glutamate excitotoxicity, mitochondrial damage, oxidative stress, and changes in neuromodulators, all of which contribute to neuronal cell death [6]. The effect of milder hypoglycemia on neurologic development is uncertain, so are the cut-offs to define and treat hypoglycemia in neonatal period [8,9,10,11,12,13,14,15]. Despite multiple guidelines, the dilemma continues [8,9,10,11,12,13,14,15]. Previous reviews suggest a potential association between early hypoglycemic episodes and adverse neurodevelopmental outcomes, including cognitive impairments and an increased risk of neurodevelopmental disorders, acknowledging the necessity for additional data to validate the effects and the lack of good-quality, large prospective studies [16,17].

We aimed to analyze the association of neonatal hypoglycemia of varying severity on long-term neurodevelopmental outcomes in the neonatal period. This is an updated analysis of the previous meta-analysis conducted by Shah et al. in 2019, including data from seven new prospective studies [17].

## 2. Methods

The systematic review was conducted according to Preferred Reporting Items for Systematic Reviews and Meta-Analyses (PRISMA) reporting guidelines [18]. Our protocol was registered on PROSPERO (CRD42024500113), the international prospective register for systematic reviews.

### 2.1. Eligibility Criteria

Randomized or quasi-randomized clinical trials and observational studies that reported any neurodevelopmental outcome following neonatal hypoglycemia were included. Studies that did not report pre-specified neonatal outcomes were excluded. Neonates with no hypoglycemic episodes were considered as comparators. 

### 2.2. Primary Outcomes

Neurodevelopmental impairment (NDI) is defined as:
Moderate to severe NDI, defined as the presence of one or more of the following: Bayley Scales of Infant and Toddler Development (BSID) III/IV [19] (cognitive, motor, or language score) <85, cerebral palsy, visual impairment (unilateral or bilateral blindness), or severe to profound hearing impairment (meeting the criteria for amplification) evaluated at 18–48 months of age.Severe NDI, defined as the presence of one or more of the following: BSID III/IV [19] (cognitive, motor or language score) <70, cerebral palsy with a Gross Motor Functional Classification Scale (GMFCS) level ≥ 3 [20], blindness (bilateral blindness with or without some functional vision in one or both eyes), or severe to profound hearing impairment (requiring cochlear implants in one/both ears or permanent hearing loss which prevents understanding of instructions) evaluated at 18–48 months of age OR any neurodevelopmental impairment as defined by authors.Cognitive impairment (6 months–21 years): Moderate (BSID-III < 85) or severe (BSID-III < 70) [19] or any cognitive impairment, defined by authors using any comparable validated tool.Visual impairment or visual-motor impairment, as defined by authors.Executive dysfunction, as defined by authors.

### 2.3. Secondary Outcome(s)

Motor and language composite scores, evaluated between 18 and 48 months of age by BSID-III/IV [19] or any validated tool.Motor impairment, evaluated between 18 and 48 months of age: moderate (BSID-III < 85) or severe (BSID-III < 70) [19] or defined by any comparable validated tool.Language impairment evaluated between 18 and 48 months of age: moderate (BSID-III < 85) or severe (BSID-III < 70) [19] or defined by any comparable validated tool.Motor function evaluated >4 years of age using any validated tool.Epilepsy as defined by authors.Cerebral Palsy (any type) evaluated clinically between 18 and 48 months of age.Neuropsychiatric or behavioral problems (autism, ADHD, ASD, or other) evaluated by any validated tool as reported by the authors.Low literacy and low numeracy, or low education achievement in mid-childhood and adolescence as defined by authors.

### 2.4. Information Sources and Search Strategy

A systematic literature search was conducted from inception until 31 March 2024, using the following databases: PubMed, CINAHL Plus with Full Text, Embase (Elsevier: Amsterdam, The Netherlands), and the Cochrane Central Register of Controlled Trials, or CENTRAL (Ovid). The references of the identified studies were also screened. No date or language restrictions or search filters were employed. Searches were run by a medical librarian (P.T and D.M) in consultation with the authors B.K and S.D. Details of the search strategy are provided in the Appendix A. Results were deduplicated using the SR-Accelerator De-duplicator tool (Version 2024).

### 2.5. Study Selection

The deduplicated results were transferred to Rayyan software (Version 2022, www.rayyan.ai, accessed on 15 January 2024) for screening. The title, abstract screening and full-text review of articles were performed independently by P.T, D.M, A.P. and R.G, in pairs. Any discrepancies were resolved by discussion and consensus with B.K and S.D. If necessary, trial authors were contacted by email correspondence to request missing data.

### 2.6. Assessment of Risk of Bias

All the included studies were assessed for methodological quality. For all the included observational studies, the risk of bias was assessed using a modified Newcastle–Ottawa scale [21] The following domains were evaluated: selection, comparability, and outcome. The a priori assumptions were that a score of >7/9 was deemed low risk, a score of 4–6/9 was deemed a moderate risk, and a score of ≤3/9 was deemed a high risk of bias. Two authors (A.P. and L.O.) conducted the risk of bias independently; conflicts were resolved after discussion and consensus. Similarly, S.D. and B.K. assessed the certainty of evidence (confidence in the estimate of effect) for each outcome based on the Grading of Recommendations Assessment, Development, and Evaluation (GRADE) framework [22]. Any discrepancies were resolved by discussion and consensus.

### 2.7. Data Synthesis

All the studies were combined and analyzed using Review Manager V.5.3 (Cochrane Collaboration, Nordic Cochrane Centre, Copenhagen, Denmark). The mean difference with 95% CI was calculated for continuous outcomes. For dichotomous outcomes, the OR with 95% CI was calculated from the data provided in the studies. Data was logit transformed. Adjusted odds ratios for potential confounders were extracted from the studies reporting these data if available. The random effects model was used to calculate summary statistics owing to the anticipated heterogeneity. The statistical heterogeneity was assessed by use of the Cochran *Q* statistic and by use of the *I*^2^ statistic, which is derived from the *Q* statistic and describes the proportion of the total variation that is due to heterogeneity beyond chance. We used the Egger regression test and funnel plots to assess publication bias if more than 10 studies were available for that particular outcome. Sensitivity analysis was performed based on the severity of blood glucose <20 mg/dL (<1.1. mmol/L), 20–34 mg/dL (1.1–1.9 mmol/L) and 35–45 mg/dL (1.9–2.5 mmol/L).

A total of 1743 articles were identified through database searching. After duplicates were removed, 1660 articles underwent title and abstract screening. 39 full-text articles were assessed. A final total of 17 non-randomized studies of intervention (19 publications) were deemed eligible to be included in the final meta-analysis.

The PRISMA flow diagram is shown in Figure 1.

## 3. Results

All the included studies were cohort studies by design. Eight studies were retrospective [23,24,25,26,27,28,29,30], and nine studies were prospective in nature [31,32,33,34,35,36,37,38,39,40,41] (Summary Table 1 and Table 2).

### 3.1. Primary Outcomes

Any neurodevelopmental impairment (NDI) as defined by authors in early childhood (2–5 years) favoured the ‘No hypoglycemia’ group involving 106,049 infants from eight studies (OR = 1.16; 95% CI = 1.11–1.43; *p* = 0.0005; *I*^2^ = 48%), reporting the outcome [23,25,26,31,32,33,36,39]. The effect size of the pooled estimate is significant for NDI even in the mid-childhood (6–11 years) group (two studies, OR 3.67; 95% CI = 1.07–12.2; *p* = 0.04; *I*^2^ = 0%) (Figure 2a,b).

Any cognitive impairment was reported in five studies [24,26,33,36,39]. A total of 5306 infants were analyzed in the outcomes. Infants with hypoglycemia had significant more risk of impaired cognitive development as compared to ‘No hypoglycemia’ (OR = 2.12; 95% CI = 1.79–2.52; *p* = <0.00001). There was significant heterogeneity noted among studies (98%) (Figure 2c). However, when mild and moderate to severe cognitive impairment was analyzed separately, there was no difference between the two groups (four studies, n = 1935, OR = 1.09; 95% CI = 0.78– 1.52) [21,24,27,28] and (four studies, n = 1935, OR = 1.26; 95% CI = 0.58–2.75) [20,22,29,35], respectively (Appendix A).

Visual-motor impairment was reported in two studies [32,33]. The hypoglycemia group of infants had a three times higher risk of impairment compared to the ‘No hypoglycemia’ group (two studies, n = 545, OR = 3.33; 95% CI = 1.14–9.72, *I*^2^ = 0%) (Figure 2d).

### 3.2. Secondary Outcomes

Executive dysfunction was reported in three studies involving 1797 infants [23,33,39]. The effect size of pooled estimated was statistically significant, favoring the ‘No Hypoglycemia’ group. The OR = 1.99; 95% CI = 1.36–2.91, *p* = 0.0004, *I*^2^ = 84%), with the caveat that the studies were widely heterogenous (Figure 2e). Low language or literacy could not be meta-analyzed, as only one study reported the outcomes [32]. Low educational achievement in mid-childhood was also reported by one study (9–10 years) [41]. The study reported no difference in any outcomes between the two groups (Appendix A).

Epilepsy, as reported by the authors, was no different between the two groups (four studies, n = 772, OR = 1.93; 95% CI = 0.78–4.80, *I*^2^ = 0%) [20,22,28,29]. Other outcomes, like hearing impairment [33] (one study, n = 456, OR = 0.85; 95% CI = 0.6–1.21), motor impairment [32,33,36,39] (four studies, n = 5502, OR = 1.48; 95% CI = 0.98–2.24, *I*^2^ = 0%), and emotional-behavioral difficulties [24,32,33] (three studies, n = 587, OR = 1.01; 95% CI = 0.73–1.39, *I*^2^ = 0%) were also reported to be similar among the two groups (Appendix A).

### 3.3. Sensitivity Analyses Based on Different Blood Glucose Cut-Offs

We performed sensitivity analyses based on different blood glucose (BG) concentrations: (a) <20 mg/dL (<1.1. mmol/L), (b) 20–34 mg/dL (1.1–1.9 mmol/L), and (c) 35–45 mg/dL (1.9–2.5 mmol/L) (Appendix A).

In severe hypoglycemia, as defined with BG <20 mg/dL (<1.1 mmol/L), NDI (2–5 years) [31,32], epilepsy and emotional-behavioral difficulties were reported for each outcome by two studies [24,32]. The effect sizes of the pooled estimates for each outcome was not statistically significant between the hypoglycemia and ‘no hypoglycemia’ groups. In infants with a BG between 20 and 34 mg/dL (1.1–1.9 mmol/L), NDI in early childhood (2–5 years) was more common in the group with hypoglycemia [25,26] (two studies, n = 1871, OR = 2.20; 95% CI = 1.21–4.02, *p* = 0.01; *I*^2^ = 0%). In the subgroup analysis of infants with BG between 35 and 45 mg/dL 1.9–2.5 mmol/L), only one study reported on the primary and secondary outcomes. There was no difference noted between the two groups [33].

### 3.4. Risk of Bias (RoB) and Certainty of Evidence (CoE)

All the non-randomized studies were assessed for quality of evidence using the Newcastle–Ottawa scale [21]. All 17 studies were adjudged as good (>6), and one study was labelled as ‘fair’ (Appendix A). The number of studies were insufficient for any outcomes to evaluate for publication bias.

Certainty of evidence for primary outcomes was adjudged ‘low’ to ‘very low’ (Table 3). Most of the outcomes were downgraded for ‘serious’ risk of bias, serious inconsistency and imprecision. Outcomes like visual motor impairment were upgraded by one level due to the large magnitude effect (Table 3).

## 4. Discussion

Low-quality evidence from large observational studies showed a significant association between hypoglycemia in the early neonatal period, and long-term neurodevelopmental problems. We confirm the previously identified associations. Additional studies with a long enough follow-up are needed to determine the cut-off concentration and to quantify the impact beyond the infancy period.

A threshold for the blood glucose concentration below which the adverse effects are seen, also referred to as a ‘relatively safe” blood glucose level, still remains a ‘conundrum’ for the neonatologist [1,42]. Globally, nearly ten guidelines exist for neonatal hypoglycemia with different target glucose concentrations and thresholds for intervention including pharmacotherapy [43]. The lack of consistency in the guidelines is intriguing since neonatal hypoglycemia is the most common preventable adverse metabolic condition in neonatal period [1], as it can lead to detrimental effects on the developing brain [5,44,45,46]. The effects are permanent and devastating not only in immediate neonatal period, with a carryover effect into the early- and mid-childhood period [17].

Our meta-analysis showed that hypoglycemia in neonatal period increased the risk of neurodevelopmental impairment by 1.5- and 3.5-fold in early- and mid-childhood, respectively. On the sensitivity analysis, NDI was more common with a BG between 20 and 34 mg/dL (1.1–1.9 mmol/L) in the early childhood group alone. This sensitivity analysis did not generate a hypothesis on which cut-off concentration affects the particular outcome. We also found that the risk of cognitive impairment and executive dysfunction doubled in infants with hypoglycemia. Visual-motor integration problems were also three times more common in hypoglycemic infants. Our study gives additional insights to the previously published meta-analyses [17]. The result of the meta-analysis comes with the caveat that confounding factors are not adjusted for most of the studies, which is its biggest drawback. So, ‘unadjusted OR’ was chosen to pool the data and to obtain meaningful results. The quality of evidence for most of the outcomes was ‘very low’. Bolyut et al. [16], in the first systematic review on hypoglycemia, had previously suggested the ideal study design to assess the long-term outcomes post-hypoglycemia. A prospective nested randomized control trial, with a minimum of 300 patients in each (sub) group, analyzing glucose by the hexokinase/glucose oxidase method, with subcategories of treatment based on glucose (<1.8, 1.8–2.6 and >2.6 mmol/L), should mean that the outcome measurement is long enough (5–7 years) to assess the function, like executive function, IQ and, more importantly, adjusting for other confounding factors. In our meta-analysis, only 1 of the 17 studies satisfied these criteria [34].

NDI and cognitive impairment are due to the overall death of cortical cells (neurons). Visual-motor integration issues are associated with the injurious effect on ventral and dorsal cortical visual streams, and executive functions suffer from the detrimental effect on the prefrontal cortex [47]. Traditionally, neonatal hypoglycemic encephalopathy, or NHBI, is associated with bilateral occipital-posterior cortical injury [5,45,48]. We know that different areas of the brain are differentially susceptible to hypoxia [45,46]. Similarly, from our findings, we interpret that certain areas of the brain could also be differentially susceptible to hypoglycemia and its sequelae. We presumed that, with increasing thresholds for hypoglycemia concentrations, different functions would be affected, which represent different areas of the brain. One possible conclusion could be to consider the different blood glucose concentrations as the indicator for different brain regions and their susceptibility to injury due to lack of glucose. Therefore, lowering the concentration to the lowest, ‘normal’ concentration appears to be a (very) dangerous approach, which may no longer enjoy equipoise by physicians and parents.

The prognosis for neonatal hypoglycemia could also vary depending on the duration of hypoglycemia; however, the evidence for this is lacking, due to the varied protocols of screening and management across different units. This makes characterization of the disease very difficult. Severe hypoglycemia can lead to long-term neurodevelopmental problems, including cognitive deficits, motor impairments, behavioral issues, and an increased risk of visual or auditory deficits [17]. There is also a risk reported of developing epilepsy later in life due to hypoglycemia-induced brain injury [49,50]. The dearth of data for the outcomes on executive function, mathematical skills, and visual-motor integration can mask the menace of hypoglycemia in the late-childhood or adolescent period, when the demand for these skills is paramount. It also mandates the long-term follow up required, as developmental assessment during infancy can fail to recognize these complex functions.

The risk factors for hypoglycemia are well known, and these may be used for a focused approach. Infants born to diabetic mothers are at higher risk due to excessive insulin production [51]. Preterm infants often have lower glycogen stores and immature liver function [51]. Newborns with intrauterine growth restriction (IUGR) have reduced energy reserves [52]. In addition, conditions such as perinatal asphyxia or hypothermia can increase glucose consumption [51]. Therefore, newborns at risk should be closely monitored for hypoglycemia, especially within the first few hours to days of life. Protocols to standardize care are useful. Breastfeeding or formula feeding, if necessary, should be initiated. The standard operating procedures must not only focus on known risk factors, but also focus on immediate treatment with oral feeding, glucose gel, or intravenous glucose, which can all prevent adverse outcomes. Breastfeeding or formula feeding should be initiated at an early stage. Infants who experience significant hypoglycemia should have regular developmental assessments to monitor any emerging issues. Unfortunately, this is very challenging and requires significant resources.

The final remark that needs to be addressed is that lack of difference between some different concentrations of glucose and the presence of NDI might be caused by the omittance of hypoglycemic incidence in the analyzed studies. There are small studies and reports that show shorter and longer hypoglycemic incidences, which could have been missed [53]. Meta-analysis on the use of CGM in newborns has shown that the sensitivity of this technology can be very high [54]. In such a fragile population as newborns, and especially premature newborns exposed to different diseases and hypoxic events, continuous glucose monitoring (CGM) should be used. The lack of standardized monitoring procedures based on CGM technology is limited in its availability in this population; nevertheless, the professional community of neonatologists and intensivists should lobby for the wider availability of this methodology to neonatal patients [55].

Our study has several strengths and weaknesses. One strength is that we analyzed many clinical studies. The underlying weakness of having limited long-term follow-up data cannot be overcome. In addition, we tried to identify different concentrations of glucose to be associated with different outcomes, which was not successful. The topic of hypoglycemia in the neonatal period is a condition which offers prevention and therapeutic interventions, which have to be consistently enforced given the identified importance in our study. In a nutshell, we have an incomplete understanding of the neurodevelopmental impact of the most frequent metabolic condition of the neonatal period [56]. More studies like CHYLD [34] need to be conducted with long term (minimum 5–7 years) follow-up to establish the causation between hypoglycemia concentration/low blood sugar concentration and the above-defined neurodevelopmental problems.

## 5. Conclusions

Asymptomatic or symptomatic neonatal hypoglycemia is strongly associated with an increased risk of neurodevelopmental impairment in early- and mid-childhood. More well-designed studies with adequate long-term follow-up are required in the future. Until then, rigorous screening of high-risk infants is necessary to prevent adverse outcomes.

## Figures and Tables

**Figure 1 life-14-01618-f001:**
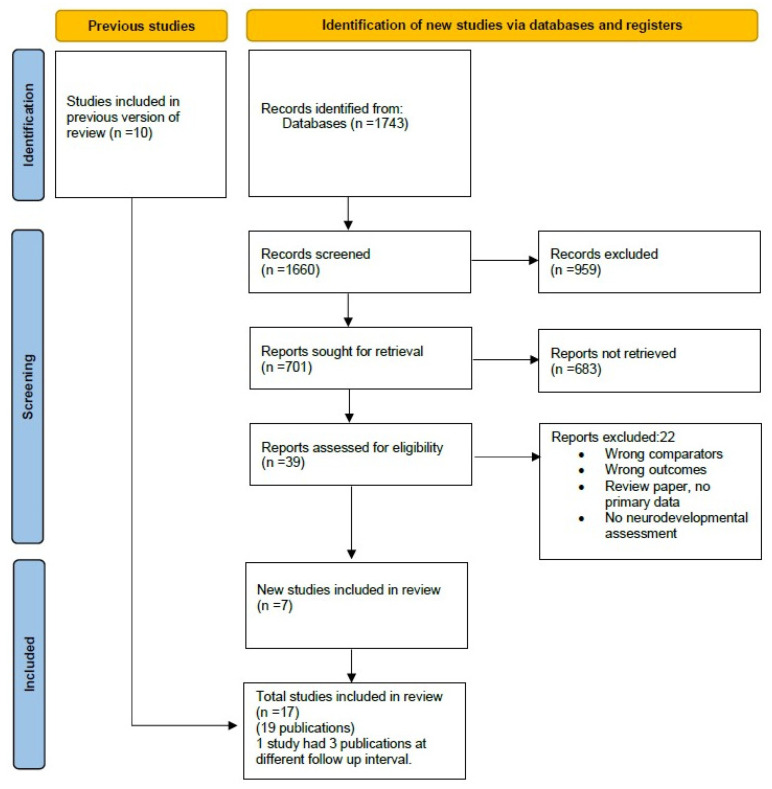
PRISMA diagram on selection of included studies.

**Figure 2 life-14-01618-f002:**
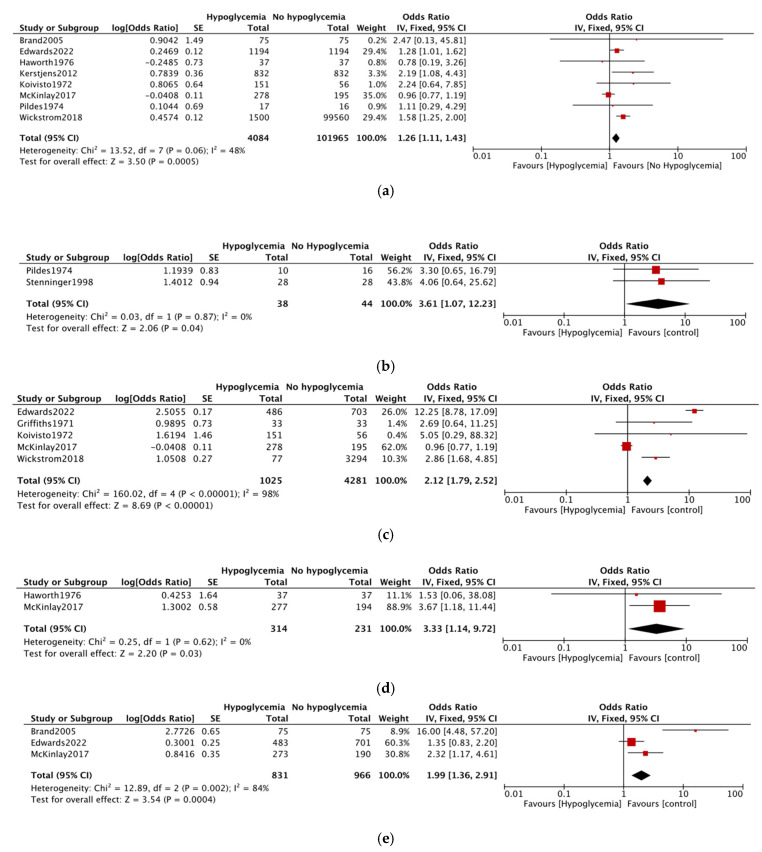
(**a**) Neurodevelopmental impairment (NDI) (early childhood) (**b**) NDI (mid-childhood) (**c**) Any cognitive impairment (**d**) visual-motor problems (**e**) executive dysfunction [23,25,26,31,32,33,36,39].

**Table 1 life-14-01618-t001:** Summary table of retrospective included studies. Study ID.

	Study Type	Inclusion Criteria	Sample Size Case vs. Controls	Asymptomatic vs. Symptomatic	Asymptomatic vs. Euglycemia	Symptomatic vs. Euglycemia	Hypoglycemia Definition	Test Method	Neurodevelopmental Outcome Assessment Age	Developmental Scale Used	Domains Assessed (Cognitive/Motor, etc.)	Confounding Factors Mentioned and Names	Methodologic Quality According to Boyut et al. [16]
Griffiths, 1971, UK [24]	Retrospective	Neonates admitted to special care unit	41 exposed, 41 unexposed	NI	NI	NI	<1.11 mmol/L (<20 mg/dL)	Capillary whole blood; modified Watson method	4.2 years mean	Cognition: Stanford-Binet or Griffiths; Behavior: Scott Systemic Interview; Motor: Griffiths Locomotor Scale; Vision: Stycar	Cognitive, behavior, motor, vision	No	Low
Koivisto, 1972, Finland [26]	Retrospective	At-risk or symptomatic neonates	151 exposed, 56 unexposed	66 asymptomatic vs. 85 symptomatic	66 asymptomatic vs. 56 euglycemia	85 symptomatic vs. 56 euglycemia	<1.7 mmol/L (<30 mg/dL)	Capillary whole blood; laboratory-modified Hultman or glucose oxidase method	1–4 years	Cognitive, language, motor tests not specified; Behavioral assessment not specified;	Cognitive, language, motor, behavioral assessment, vision	No	Low
Stenninger, 1998, Sweden [27]	Retrospective	Infants of diabetic mothers	13 exposed, 15 unexposed	NI	NI	NI	<1.5 mmol/L (<27 mg/dL)	Capillary whole blood; laboratory glucose oxidase method	7–8 years	Cognitive: Griffiths Developmental Scale; Motor: Movement Assessment Battery for Children; Behavior questionnaires not specified; Validated neurological screening test for evaluation of minimal brain dysfunction; Electroencephalogram	NI	No	Medium
Duvanel, 1999, Switzerland [28]	Retrospective	Preterm ≤ 34 weeks and small for gestational age	85 infants; 62 exposed, 23 unexposed	NI	NI	NI	<2.6 mmol/L (<47 mg/dL)	Dextrostix for screening; if <2 mmol/L confirmed by laboratory venous sample using glucose oxidase or hexokinase method	6, 12, 18 months and 3.5 and 5 years	Cognitive: Griffith’s Developmental Scales, McCarthy Scales of Aptitude	Cognitive, development	No	Low
Brand, 2005, Netherlands [23]	Retrospective	Term and large for gestational age	75 infants; 60 exposed, 15 unexposed	NI	NI	NI	<2.2 mmol/L (<40 mg/dL)	Capillary whole blood; laboratory glucose oxidase method	4 years	Denver Developmental Scale, Snijders-Oomen non-verbal intelligence test, Dutch version of the Child Behavior Check List	Cognitive, behavior	No	Low
Kerstjens, 2012, Netherlands [25]	Retrospective	Moderate preterm (32–35 weeks gestation)	832 infants; 67 exposed, 765 unexposed	NI	NI	NI	<1.7 mmol/L (<30 mg/dL)	Bedside glucometer; laboratory confirmation if <3 mmol/L (54 md/dL) or <2.5 mmol/L (45 mg/dL), depending on site protocol	3.5–4 years	Ages and Stages Questionnaire	NI	Yes	Medium
Kaiser, 2015, USA [29]	Retrospective	All neonates	1395 infants; 89 exposed, 1306 unexposed	NI	NI	NI	<1.94 mmol/L (<35 mg/dL)	Laboratory glucose oxidase	10 years	Forth Grade Benchmark Examination	Literacy and mathematics	Yes	Medium
Goode, 2016, USA [30]	Retrospective	Preterm and low birth weight	743 infants; 461 exposed, 282 unexposed	NI	NI	NI	<2.49 mmol/L (<45 mg/dL)	Dextrostix or plasma glucose; method not specified	5, 8, and 18 years	Cognitive: Stanford-Binet, Peabody Picture Vocabulary Test, Wechsler Intelligence Scale for Children, Wechsler Abbreviated Scale of Intelligence. Academic: Woodcock Johnson Test of Achivement. Behavior: Child Behavior Checklist, Youth Report Behavior Surveillance System	Cognitive, academic, behavior	Yes	Low

Legend: NI = No information available.

**Table 2 life-14-01618-t002:** Summary table of prospective studies included.

Study ID	Study Type	Inclusion Criteria	Sample Size Case vs. Controls	Asymptomatic vs. Symptomatic	Asymptomatic vs. Euglycemia	Symptomatic vs. Euglycemia	Hypoglycemia Definition	Test Method	Neurodevelopmental Outcome Assessment Age	Developmental Scale Used	Domains Assessed (Cognitive/Motor, etc.)	Confounding Factors Mentioned and Names	Methodologic Quality According to Boyut et al. [16]
Pildes, 1974, USA [31]	Prospective	At-risk or symptomatic neonates (mostly preterm or SGA)	39 exposed, 41 unexposed	NI	NI	NI	<1.11 mmol/L (<20 mg/dL)	Capillary whole blood; laboratory glucose oxidase method	1–7 years	Cognitive: Cattell Infants Scale, Stanford-Binet or Wechsler Intelligence Scale for Children; Social: Vineland Social Maturity Scale	Cognitive, social, electroencephalogram	No	Low
Haworth, 1976, USA [32]	Prospective	Infants of diabetic mothers	25 exposed, 12 unexposed	NI	NI	NI	≤1.11 mmol/L (≤20 mg/dL) in low birthweight babies (<2.5 kg) and ≤1.65 mmol/L (≤30 mg/dL) in normal weight babies	Capillary whole blood; laboratory Huggett and Nixon method	4.5 years mean	Yale Developmental Schedule	NI	No	Low
McKinlay, 2015, New Zealand [34]	Prospective	Term and late-preterm neonates at risk for hypoglycemia	404 infants; 216 exposed, 188 unexposed	NI	NI	NI	<2.6 mmol/L (<47 mg/dL)	Capillary whole blood laboratory glucose oxidase method; masked continuous interstitial glucose monitoring	2 years	Cognitive: Bayley Scales of Infant Development III. Executive: a battery of four tasks, and Behavior Rating Inventory of Executive Function (preschool version) Vision: visual screening using four assessment categories and random-dot kinematograms of varying coherence. Hearing: audiologic screening	Cognitive, Executive, Vision, Hearing	Yes	Medium
Mahajan, 2017, India [35]	Prospective	Infants of gestational age greater than 32 weeks who developed hypoglycemia during the first seven days of life	142 infants; 72 hypoglycemia, 70 euglycemia	NI	NI	NI	Blood glucose levels less than 50 mg/dL	Glucose test strips, confirmed by laboratory plasma glucose (hexokinase method)	6 months and 12 months	Neurological assessment was performed by using the Amiel-Tison Scale, 10 and developmental assessment was done by using the Developmental Assessment Score for Indian Infants (DASII) scoring system	NI	Yes	Medium
McKinlay, 2017, New Zealand [33]	Prospective	Term and moderate to late preterm infants born at risk of hypoglycemia	477 infants; 280 exposed, 197 unexposed	NI	NI	NI	<2.6 mmol/L (<47 mg/dL)	Capillary whole blood laboratory glucose oxidase method; masked continuous interstitial glucose monitoring	4.5 years	Cognitive: Wechsler Preschool and Primary Scale of Intelligence (version 3). Executive: a battery of five graded tasks, Behavior Rating Inventory of Executive Function. Motor: Movement Assessment Battery for Children (version 2) and Beery Buktenica Developmental Test of Visual Motor Integration (version 6). Vision: visual screening using six assessment categories, visual processing subscale of BBV-MI6, random dot kinetograms of varying coherence. Auditory processing: auditory subscale of the Phelps Kindergarten Readiness Scale. Emotional and behavior: Strengths and Difficulties Questionnaire, Child Behavior Checklist	Cognitive, Executive, Motor, Vision, Auditory processing, Emotional, and Behavior	Yes	Medium
Wickstrom, 2018, Sweden [36]	Prospective	All singletons born 1 July 2008 through 31 December 2012 having a diagnosis of transitory neonatal hypoglycemia	101,060 infants; 1500 hypoglycemia, 99,560 no hypoglycemia	NI	NI	NI	Blood glucose <40 mg/dL within 6 h after birth	Bedside analyses	2–6 years	NI	Any developmental delay; motor developmental delay; and cognitive developmental delay	Yes	Medium
Qiao, 2019, China [37]	Prospective	Infants born from diabetic mothers	339 infants; 195 exposed, 144 unexposed	NI	NI	NI	At least one episode of blood glucose concentration (BGC) less than 2.6 mM within 0.5 h after birth	Microglucose meter	2 years	Gesell Developmental test (Chinese revised version)	Gross motor skills, fine motor skills, adaptability, language, adaptability, social skills	No	Low
Rasmussen, 2020, Denmark [38]	Prospective	One or more episodes of moderate or severe hypoglycemia	103 children; 71 hypoglycemia, 32 healthy siblings in the control group	NI	NI	NI	Blood glucose before 2 h of age between 0.5 and 1.0 mmol/L (0–18 mg/dL; moderate), or below 0.5 mmol/L (9 mg/dL; severe); and from 2 h onwards 1.0–1.6 mmol/L (18–29 mg/dL; moderate), or below 1.0 mmol/L (severe)	Blood glucose	6–9 years	Wechsler’s intelligence scale for children fourth edition (WISC-IV) [13], Movement Assessment Battery for Children 2 (Movement ABC-2), Achenbach Child Behavior Checklist (CBCL)	Cognitive function, motor function and behavior	Yes	Low
Edwards, 2022, New Zealand [39]	Prospective	Newborns less than 1 h after birth, one or more risk factors for hypoglycemia (maternal diabetes, small [birthweight <2.5 kg or <10th centile], large [birthweight >4.5 kg or >90th centile], or preterm [35–36 weeks’ gestation]), birthweight greater than or equal to 2.2 kg, gestation greater than or equal to 35 weeks	1194 infants; 704, normoglycemia group, 490, hypoglycemia group	NI	NI	NI	One or more episodes of consecutive blood glucose concentrations less than 47 mg/dL in the first 48 h after birth	Glucose oxidase method, 2 h after birth	2 years	Cognitive: Bayley Scales of Infant and Toddler Development, Third Edition (Bayley-III) cognitive, language, and motor scales (mean [SD], 100 [15]) composite score	Blindness (visual acuity 1.3 logMAR), Hearing impairment requiring aids, cerebral palsy, developmental delay (Bayley-III cognitive, language, or motor composite score < 85), or performance-based executive function total score more than 1.5 DS below the cohort mean	Yes	High
Kennedy, 2022, New Zealand [40]	Prospective	Children born at or after 32 weeks’ gestation with one or more risk factors for neonatal hypoglycemia: maternal diabetes, preterm birth (90th percentile or >4500 g) birthweight, or other illness	99 infants; 31 normoglycemia group, 68 hypoglycemia group	NI	NI	NI	At least one consecutive blood glucose concentration <2.6 mmol/L and an interstitial episode was defined as at least 10 min of interstitial glucose concentrations <2.6 mmol/L.	NI	9–10 years	Executive function was assessed using a tablet-based battery (Cambridge Neuropsychological Test Automated Battery). Measures of attention (Attention Switching Task [AST]), planning/memory (one-touch stockings of Cambridge, Spatial Working Memory [SWM], Paired Associate Learning [PAL]) and inhibition (Stop Signal Task [SST]). Academic achievement was assessed using the e-asTTle school achievement tests in reading comprehension and mathematics in English or Māori. Social-emotional behavior was assessed using the strengths and difficulties questionnaire (SDQ)	Executive function, academic achievement, and social-emotional behavior	Yes	Low
Shah 2022 [41]	Prospective	Term and moderate to late preterm infants born at risk of hypoglycemia	587 infants; 280 exposed, 197 unexposed	NI	NI	NI	<2.6 mmol/L (<47 mg/dL)	Capillary whole blood laboratory glucose oxidase method; masked continuous interstitial glucose monitoring	9–10 years	Academic achievement was assessed in English or Māori using a standardized curriculum-based online achievement test, the Electronic Assessment Tools for Teaching and Learning (e-asTTle)	Academic achievement, executive function, visual-motor function, psychosocial adaptation, and general health	Yes	Medium

Legend: NI = No information available.

**Table 3 life-14-01618-t003:** GRADE table for important outcomes.

Outcomes	Exposure Effect (95% CI)	№ of Participants (Studies)	Certainty of the Evidence(GRADE)
Neurodevelopment impairment (<5 years)	OR 1.26 (1.11 to 1.43)	106,049 (8 non-randomized studies)	⨁◯◯◯ Very low ^a^
Neurodevelopmental impairment (>5 years)	OR 3.61 (1.07 to 12.23)	82 (2 non-randomized studies)	⨁◯◯◯ Very low ^b^
Visual motor impairment	OR 3.33 (1.14 to 9.72)	545 (2 non-randomized studies)	⨁⨁◯◯ Low ^c^
Executive dysfunction	OR 1.99 (1.36 to 2.91)	1797 (3 non-randomized studies)	⨁◯◯◯ Very low ^d^
Any cognitive impairment	OR 2.12 (1.79 to 2.52)	5306 (5 non-randomized studies)	⨁◯◯◯ Very low ^e^

Explanation: ^a^ Downgraded by three levels for high risk of bias, heterogeneity, ^b^ Downgraded three levels due to risk of bias and imprecision. Upgraded one level for a very large effect. ^c^ Downgraded three levels due to risk of bias and imprecision. Upgraded one level for a very large effect. ^d^ Downgraded three levels due to risk of bias and imprecision and inconsistency. ^e^ Downgraded three levels due to the risk of bias and imprecision.

## Data Availability

Published data is already available in public domain from all included studies. No new data were created or analyzed in this study.

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
