# Peer review of "Neonatal Hypoglycemia and Neurodevelopmental Outcomes—An Updated Systematic Review and Meta-Analysis"

_life, 2024, doi:10.3390/life14121618_

Round 1

Reviewer 1 Report

Comments and Suggestions for Authors

This review, in my opinion, is highly fascinating and crucial to the current clinical understanding of neonatology. The authors analysed an enormous number of studies devoted to neoanatal hypoglycemia, the very important and common complication among newborns.

The authors confirmed that asymptomatic or symptomatic neonatal hypoglycemia is strongly associated with an increased risk of neurodevelopmental impairment in early and mid-childhood. In my opinion, this review fulfilled all criteria to be published

Author Response

Thank you for your valuable feedback.

Please see the attachment below.

Reviewer 2 Report

Comments and Suggestions for Authors

I read with interest the manuscript entitled "Neonatal Hypoglycemia and Neurodevelopmental outcomes – An updated systematic review and metanalysis".

You are missing keywords.

Within the introduction, you repeat sentences and some references are not listed in accordance with the instructions for authors. Please explain and present the previous systematic reviews on the above topic in more detail. The aim is clearly stated.

The sentence "The following outcomes were included in this meta-analysis: " is unnecessary. Remove it.

The materials and methods section is generally well written so that the study could be reproducible.

It is more logical for figure 1 to be in front of table 1.

Please use PRISMA 2020 flow diagram for updated systematic reviews which included searches of databases, registers and other sources! https://www.prisma-statement.org/prisma-2020-flow-diagram

The results section is correctly presented.

You must start the discussion by presenting your most relevant results in knowledge, which you must then compare with previous knowledge.

Define the strengths and weaknesses of your systematic review more specifically at the end of the discussion.

According to the template at the end of the manuscript, remove the following "What's Known on This Subject" and "What This Study Adds". Add "Supplementary Materials:", "Author Contributions:", "Funding:", "Data Availability Statement:", "Acknowledgments:".

Author Response

(The authors gave the same response as above.)

Reviewer 3 Report

Comments and Suggestions for Authors

Thank you very much for allowing me to review the systematic review manuscript titled “life-3289338_Neonatal Hypoglycemia and Neurodevelopmental outcomes – An updated systematic review and metanalysis. “, submitted for the section “Physiology and Pathology “. The aim of the study was to perform an updated meta-analysis on neonatal hypoglycemia, the severity of hypoglycemia, and its associated neurodevelopmental outcomes from infancy to adulthood.

Summary: When specifying the time period reviewed, the start and end dates should be clearly and concisely indicated. The first time the acronym NDI is used, its meaning (Neurodevelopmental Impairment) must be stated. When mentioning the parameter risk from line 28, it should be clarified: Neurodevelopmental problems?

Introduction: Given the importance of the topic, this section should be expanded to include more information on the prevalence of neonatal hypoglycemia, the pathophysiological mechanisms that can lead to neurological damage, and an extended review of previous studies (lines 55–58). This should evaluate the knowledge these studies contributed as well as the weaknesses that this analysis aims to address. Additionally, stating that this study is an extension of a previous one is redundant, as it has already been mentioned earlier.

Methodology: In section 2, where the scales used are listed, these scales should be properly referenced (e.g., lines 75, 80, etc.). In subsection 2.4, not only the end date of the article search period but also the precise start date should be provided. The article selection diagram should be included. The PRISMA article selection figure belongs in the methodology section, not in the results, and it should be corrected. For instance, each database used in the search should have a separate box in the identification phase. Excluded records should be justified in more detail.

Results: The results are presented in tables that enhance clarity. However, these tables should be organized chronologically, from the oldest to the most recent studies. Table 1 is excessively large and should be divided, for example, into prospective and retrospective studies. Additionally, a column indicating whether the study found an association between hypoxia and neurological impairment in children should be included.
Figure 2 requires improvement in clarity, as it is currently difficult to read due to a lack of sharpness.

Discussion : The discussion should begin by addressing the study objective. However, the authors start by discussing glucose levels, when the focus should be on whether hypoglycaemia affects a child’s neurodevelopment. While the discussion is compelling, it should also highlight the strengths and weaknesses of the studies reviewed, as well as the need for future studies to address the limitations identified in this meta-analysis.

The conclusion is consistent with the results obtained.

Author Response

Thank you for your valuable feedback.
